# Inter-Observer Agreement in Radiographic Diagnosis of Coxofemoral Joint Disease in a Closed Cohort of Four-Month-Old Rottweilers

**DOI:** 10.3390/ani12101269

**Published:** 2022-05-15

**Authors:** Britta Vidoni, Masoud Aghapour, Sibylle Kneissl, Aldo Vezzoni, Michaela Gumpenberger, Harald Hechinger, Alexander Tichy, Barbara Bockstahler

**Affiliations:** 1Small Animal Surgery, Department for Companion Animals and Horses, University of Veterinary Medicine, 1210 Vienna, Austria; britta.vidoni@vetmeduni.ac.at; 2Section of Physical Therapy, Small Animal Surgery, Department for Companion Animals and Horses, University of Veterinary Medicine, 1210 Vienna, Austria; barbara.bockstahler@vetmeduni.ac.at; 3Diagnostic Imaging, Department for Companion Animals and Horses, University of Veterinary Medicine, 1210 Vienna, Austria; sibylle.kneissl@vetmeduni.ac.at (S.K.); michaela.gumpenberger@vetmeduni.ac.at (M.G.); hechinger@me.com (H.H.); 4Clinica Veterinaria Vezzoni S.R.L., 26100 Cremona, Italy; clinica@vezzoni.it; 5Platform Bioinformatics and Biostatistics, Department for Biomedical Services, University of Veterinary Medicine, 1210 Vienna, Austria; alexander.tichy@vetmeduni.ac.at

**Keywords:** hip laxity, canine hip dysplasia, inter-observer agreement, radiographic examination, Rottweiler

## Abstract

**Simple Summary:**

Canine hip dysplasia is an important orthopedic disorder in veterinary medicine. In addition to body weight, fat intake, rapid growth, and hormonal changes, heredity is one of its underlying factors. Diagnosis of hip dysplasia under one year of age would help veterinarians to plan proper preventive/therapeutic methods and exclude dysplastic dogs from breeding programs to reduce the incidence of the disease in future generations. Therefore, having an accurate method for the early diagnosis of dysplastic dogs is an important subject, and a limited number of screening methods are used globally. Assessment of the accuracy of these methods can be examined with intra- and inter-observer studies. In this study, we aimed to assess the inter-observer agreements of the eight radiographic parameters from four different radiographic projections between five experienced observers (two of them certified scrutineers for canine hip dysplasia) to evaluate the reproducibility of the protocols. In our study, high inter-observer agreements were recorded for measurable values such as angle measurements, whereas the results of the subjective grading were low. Thus, it can be deduced that quantitative parameters are reliable values and a combination of these values with other screening programs such as clinical examinations might increase the accuracy of the examinations.

**Abstract:**

Canine hip dysplasia is a complex and multifactorial disease. The early diagnosis of dysplastic dogs under one year of age helps veterinarians to plan proper preventive/therapeutic methods. Having an accurate screening method increases the chance of the early detection of dysplasia. The goal of our study was to assess the inter-observer reliability of eight radiographic parameters in four-month-old Rottweilers. Radiographs of the 28 Rottweilers were investigated by five experienced observers. The radiographs were taken in ventrodorsal view with extended legs, frog-leg ventrodorsal view, distraction view, and dorsal acetabular rim view. Four quantitative parameters such as Norberg angle (NA), distraction index (DI), dorsal acetabular rim slope (DARS), and center edge angle (CEA) and four qualitative parameters such as sclerosis of the cranial acetabular rim (SCAR), location of the center of the femoral head (LCFH), grading of the degenerative joint disease (GDJD), and grading of the dorsal acetabular rim (GDAR) were evaluated. High inter-observer agreements were recorded for quantitative values, whereas the inter-observer agreement of the qualitative parameters was low. It can be deduced that the evaluated quantitative parameters are reliable, and a combination of these methods with clinical examinations might increase the accuracy of the examinations.

## 1. Introduction

Canine hip dysplasia (CHD) is a progressive, complex, and polygenic disease, which develops during postnatal skeletal growth and is characterized by malformation of the coxofemoral joint leading to joint laxity, subsequent degenerative joint disease, and loss of articular function [1,2]. The main etiology of CHD is still under investigation, but genetics, dog breed, body weight and size, rapid growth, fat intake, and hormonal changes have been reported to be the most important factors [1,2,3,4,5,6]. Hip laxity is reported to be a primitive sign of CHD and an important predisposing factor of degenerative joint diseases [7,8]. A direct relationship has been reported between the severity of coxofemoral laxity and the incidence of degenerative joint disease [7,8]. Ossification disorders of the acetabulum and degeneration of the femoral head ligament and joint capsule have been considered as the main causes of hip laxity [9,10]. Hip laxity causes micro-fractures in the subchondral bone during the gait cycle, resulting in osteoarthritis [9].

Early diagnosis of hip laxity in medium- to large-breed dogs for a screening time of one year would make veterinarians capable of determining predisposed or affected hips at young ages and performing preventive or therapeutic procedures to restrict the disease or reduce the symptoms of CHD in older ages. These procedures would promote the animals’ welfare and reduce therapeutic costs in the future. Furthermore, dysplastic dogs should be excluded from breeding programs due to the importance of heredity in this disease (OMIA 000473-9615) [3,4,11]. Breeding is multifactorial, and excluding solely CHDs would reduce the genetic diversity of a breed especially those with historically increased laxity.

Early diagnosis of CHD is based on clinical orthopedic examinations and different radiographic screening methods. Clinical orthopedic methods consist of qualitative methods such as Barlow, Ortolani, and Bardens tests which are based on the examination of the instability of the hip joint [12,13,14], and quantitative examinations such as measurements of the reduction or subluxation angles [15].

The Ortolani test can be performed from 7 to 8 weeks of age [16] but a high number of false-negative results (up to 85%) can be seen at this age [17]. The results of the Ortolani test, from 16 weeks of age, were reported to be more accurate, with 92% sensitivity [17,18]. The Bardens test can be conducted from 8 to 9 weeks of age with 83% accuracy [19]. It should be considered that the degree of hip-joint laxity can be raised between the ages of 8 and 12 weeks due to the expansion of the joint capsule and false-positive results might be observed [20,21]. On the other hand, in juvenile dogs with severe grades of hip laxity, false-negative orthopedic examinations might be recorded because of the fibrosis of the joint capsule [20]. Thus, due to the possibility of false-negative or false-positive results, a combination of the physical and radiographic examinations is recommended [22,23,24].

Along with physical examinations, radiography is widely used for screening for CHD. The main aim of the radiographic examination is to determine hip-joint laxity or signs of degenerative joint diseases. Limited numbers of standardized radiographic screening protocols, such as guidelines of the Fédération Cynologique Internationale (FCI), the Pennsylvania Hip Improvement Program (PennHIP), the Orthopedic Foundation for Animals (OFA), and the British Veterinary Association/The Kennel Club (BVA/KC) are used globally [25,26,27,28].

In a previous study, we focused on clinical orthopedic examinations for early diagnosis of hip laxity and investigated the correlation between clinical examination and final radiographic score according to the FCI guidelines. A positive correlation was recorded between orthopedic examinations (reduction angle and Ortolani maneuver) and final FCI score from 8 months of age in that study [23].

Assessment of the objectivity of radiographic studies can be made by agreement studies. These studies aim to distinguish the methods with no-to-good agreements. Since the interpretation of the radiographs relies on the observer, in the case of low agreements, the results may not represent the investigated values and lead to inaccurate data. Thus, the recommended imaging method must have good-to-high agreements. Given that the radiographic landmarks might be less distinct or even missing for identifying and scoring coxofemoral joint disease in juvenile dogs, the main objective of the current study was to investigate the inter-observer agreement of the eight radiographic parameters in a closed cohort of four-month-old Rottweilers. We hypothesized that there would be a good inter-observer agreement for the values measured in this study, due to the experience of the observers.

## 2. Materials and Methods

### 2.1. Approval and Consent

This study was discussed and approved by the Institutional Ethics and Animal Welfare Committee of the University of Veterinary Medicine, Vienna in accordance with good scientific practice guidelines and national legislation (ETK-17/12/97/2015).

### 2.2. Study Design and Inclusion Criteria

In this prospective study, a closed cohort of Rottweilers was investigated. Closed cohort studies are longitudinal follow-up studies with fixed subjects driven by researchers to assess the outcomes among the groups [29]. In closed cohort studies, the subjects are fixed; thus, no new patients are added to the study, and the study population may decrease because of the loss of the subjects. A total of 28 purebred Rottweilers of the Austrian Armed Forces was investigated in our study. The dogs were representative of a defined population and all of them had different parents. All of the dogs underwent general clinical examination prior to the study. The inclusion criterion was being free of any clinical musculoskeletal diseases. The age of each dog was controlled by breeding documents and the body weight was recorded. All of the clinical examinations were performed by an experienced orthopedic surgeon (B.V.).

### 2.3. Anesthesia

Due to the importance of the radiographic positioning, and to avoid musculature contraction during the radiographic examination, all the dogs underwent general anesthesia in this study. The dogs were premedicated with medetomidine (0.01–0.02 mg/kg, IV) and inducted with propofol (1–5 mg/kg, IV). The maintenance of the anesthesia was performed with propofol (0.2 mg/kg/min, IV).

### 2.4. Radiographic Examinations

All radiographs included in this study were taken at the Diagnostic Imaging of the University of Veterinary Medicine, Vienna. The radiographs were carried out in four different projections such as ventrodorsal view with extended legs [30], ventrodorsal frog-leg view [31], ventrodorsal distraction view (Badertscher method modified by Vezzoni) [32], and dorsal acetabular rim (DAR) view [33]. The radiographs were exposed with 75–96 kV and 9 mAs with a film-focus distance of 90 cm in a storage phosphor screen/cassette system (Kodak Carestream, Health Inc., Rochester, NY, USA). All images were digitally stored and evaluated (dicomPACS View Version 6.0.2, 457, Bodmin, England and Oehm und Rehbein, Rostock, Germany). The radiographs and their orders were anonymized prior to the study by an experienced technician. Five experienced investigators evaluated the radiographs in this study, including two experienced orthopedic surgeons (observers 1 and 2) and three experienced radiologists (observers 3, 4, and 5). One of the surgeons (observer 1) and one of the radiologists (observer 4) were certified CHD scrutineers. The investigators recorded their observations for each parameter of the hips separately. The inter-observer variability was based on the measurement of the values among the five observers.

### 2.5. Investigated Parameters

Eight radiographic parameters were investigated in this study. Three parameters were evaluated in ventrodorsal view radiographs with extended legs, including Norberg angle (NA), sclerosis of the cranial acetabular rim (SCAR), and location of the center of the femoral head (LCFH). Grading of the degenerative joint disease (GDJD) was performed on frog-leg view radiographs and the distraction index (DI) was measured on distraction view radiographs according to the PennHIP method. Furthermore, three parameters were investigated in DAR-view radiographs, including dorsal acetabular rim slope (DARS), center edge angle (CEA), and grading of the dorsal acetabular rim (GDAR). These eight parameters were divided into quantitative (NA, DI, DARS, and CEA) and qualitative (SCAR, LCFH, GDJD, and GDAR) parameters.

#### 2.5.1. The Norberg Angle (NA)

The NA presents information about the location of the center of the femoral head relative to the craniolateral acetabular margin and is reported to be ≥105° in normal hips in most breeds [1]. Smaller NA values usually represent subluxated/luxated femoral heads. The measurement of the NA is based on the measurement of the angle between the straight line connecting the centers of the contralateral femoral heads in ventrodorsal radiographs and the line connecting the center of the femoral head at each side with the intersection of the dorsal rim and craniolateral border of the acetabulum. The measurement of the NA is shown in Figure 1.

#### 2.5.2. Sclerosis of the Cranial Acetabular Rim (SCAR)

The SCAR was investigated according to its uniformity and thickness and was categorized as regular and thin sclerosis (normal), regular and thick sclerosis, or laterally increased thickness of sclerosis (abnormal).

#### 2.5.3. Location of the Center of the Femoral Head (LCFH)

The LCFH was evaluated according to its position relative to the dorsal acetabular rim (medial, on, or lateral). The results were categorized as medial to DAR, superimposing DAR, 0–2 mm lateral to DAR, and more than 2 mm lateral to DAR (Figure 1).

#### 2.5.4. Grading of the Degenerative Joint Disease (GDJD)

The grading of the DJD of the coxofemoral joint was based on the evaluation of the joint space and the existence of the osteophytes. The results were classified as hips with congruent joint space and no osteophytes (normal), hips with incongruent joint space and no osteophytes, and hips with incongruent joint space with osteophytes.

#### 2.5.5. The Distraction Index (DI)

To quantify the DI, ventrodorsal distraction view radiographs (Badertscher method modified by Vezzoni) were taken, and the calculation of the DI was performed using the PennHIP measurement method [26]. The DI is calculated by dividing the distance between the centers of the femoral head and acetabulum by the radius of the femoral head during distraction and is a number between zero (healthy hip) and one (severe laxity). The incidence of hip laxity in dogs with DI ≤ 0.3 and DI ≥ 0.6 is reported to be low and high, respectively [34]. The measurement of the DI on the distraction view radiograph is shown in Figure 2.

#### 2.5.6. Dorsal Acetabular Rim Slope (DARS)

The DARS is the angle between the intersection of the perpendicular line drawn from the center of the femoral head to the mid-sagittal plane of the pelvis/sacrum, and the line drawn tangent to the most lateral point of the dorsal acetabular rim [33]. The DARS is reported to be ≤7.5° in healthy hips, whereas higher values indicate hips with increased laxity [33]. The measurement of the DARS on a DAR-view radiograph is shown in Figure 3A.

#### 2.5.7. The Center Edge Angle (CEA)

The CEA is an angle between the straight line, drawn from the center of the femoral head tangential to the outer edge of the acetabular rim, and the straight line, drawn parallel to the mid-sagittal axis of the pelvis/sacrum on DAR-view radiographs [35,36]. The CEA is used to assess the acetabular coverage of the head of the femur and is reported to be greater than 12° for healthy hips [36]. The magnification of the measurement of the CEA in a Rottweiler dog is shown in Figure 3B.

#### 2.5.8. Grading of the Dorsal Acetabular Rim (GDAR)

The GDAR was based on the shape of the DAR and degenerative changes. The results were graded into five groups such as DAR with a triangular shape and no osteophytes (group 0), mild rounded DAR and no osteophytes (group 1), rounded DAR with mild osteophytes (group 2), truncated-shape DAR where the edge looks cut off with significant laxity and poor roofing (group 3), and the rounded-shape DAR with severe osteophytes (group 4).

### 2.6. Statistical Analysis

The statistical analysis of the measured data was performed using IBM SPSS Statistics software version 25 and descriptive statistics were calculated. The differences between observers were analyzed using a general linear model (GLM) and multiple comparisons were performed using Bonferroni’s alpha correction procedure. The results with a *p*-value < 0.05 were considered significant.

The intra-class correlation coefficient (ICC) was calculated to assess the inter-observer reliabilities of the quantitative data. The ICC ranged from 0 (no agreement) to 1 (excellent agreement). The values below 0.4 were considered poor agreement, the values between 0.4 and 0.59 were considered fair agreement, an ICC between 0.6 and 0.74 was considered good agreement, and an ICC higher than 0.75 was considered excellent agreement [37].

To evaluate the inter-observer agreement of the qualitative data, Cohen’s kappa coefficient (κ) was calculated (κ = 0: no agreement, and κ = 1: excellent agreement). The values with a kappa below 0.20 indicate poor agreement, and the values between 0.21 and 0.40 indicate weak agreement. The values between 0.41 and 0.60, and between 0.61 and 0.80 indicate moderate and good agreements, respectively. Results with a kappa above 0.80 indicate excellent agreement.

A few qualitative data were reported to be indefinable by observer 5. The absence of these data might have a negligible impact on statistics.

## 3. Results

Altogether, 28 purebred Rottweilers (in total 56 hips) with a mean ± standard deviation (SD) age of 19.3 ± 1.4 weeks (range: 20.8–16.4 weeks) and a mean ± SD body weight of 16.5 ± 5.6 kg (range: 20.5–12.5 kg) were investigated in this study. Of the examined dogs, 20 were male (all intact) and 8 were female (all intact).

### 3.1. Quantitative Results

No significant difference was recorded between observers regarding NA, and the highest mean difference ± standard error (SE) between observers was 3.7° ± 1.4°. As with NA, no significant difference was recorded for DI between observers, and the highest mean difference ± SE between observers was 0.05 ± 0.03 mm.

Two of the radiologists (observers 3 and 5) who had less experience than the others measured significantly different DARS than other observers (*p*-value < 0.02) with a highest mean difference ± SE of 7.1° ± 0.9°. No significant difference was recorded between other observers (observers 1, 2, and 4) regarding DARS.

Except for one case, the measurements of the CEA did not show a significant difference between observers, either. A significant difference (*p*-value = 0.04) was only seen between observers 2 (surgeon) and 4 (radiologist) regarding CEA. The highest mean difference ± SE of the CEA between these observers was 2.9° ± 1°. The descriptive statistics of the quantitative values are summarized in Table 1. Furthermore, the results of the general linear model and Bonferroni’s alpha correction procedure are shown in the Appendix A (Appendix A).

### 3.2. Qualitative Results

From the total SCAR measurements, 84.1% of the hips were graded with regular and thin sclerosis, 12.6% of the hips were marked as regular and thick sclerosis, and 3.3% of the hips were graded as laterally increased thickness of sclerosis.

From the total LCFH measurements, 49.3% of the femoral heads were classified as medial to DAR, 29.1% were classified as superimposing DAR, 18.7% of the results were classified as 0–2 mm lateral to DAR, and 3% of the femoral heads were categorized as more than 2 mm lateral to DAR.

Evaluation of the GDJD was based on the assessment of the joint space and existence of the osteophytes and 68.1% of the hips were categorized as hips with congruent joint space and no osteophytes. Furthermore, 26.8% of the hips were categorized as hips with incongruent joint space and no osteophytes, and 5.1% were classified as hips with incongruent joint space with osteophytes.

The hips were divided into five groups based on the shape of the DAR and the existence of the osteophytes. Of all the investigated hips, 31.6% were diagnosed with a triangular shape and no osteophytes, 40.7% were diagnosed with mild rounded DAR and no osteophytes, 19.3% were diagnosed with rounded DAR with mild osteophytes, 6.2% were diagnosed with the truncated shape of the DAR where the edge looks cut off or dabbed with clear laxity and less roofing, and 2.2% of the results were categorized as a rounded shape of DAR with severe osteophytes. The results of the evaluation of the qualitative values are shown in Table 2 for each observer.

### 3.3. Inter-Observer Agreements of Quantitative Values

The ICCs recorded for NA, DI, and CEA were higher than 0.75, indicating an excellent inter-observer agreement. However, the ICC recorded for DARS was 0.74, indicating a good inter-observer agreement. The inter-observer ICCs are shown in Table 3 for each parameter.

### 3.4. Inter-Observer Agreements of Qualitative Values

Despite quantitative values, low inter-observer agreements were recorded for qualitative values. A high number (80%) of the recorded kappa values for SCAR were lower than 0.2, indicating poor-to-no agreements and only one (10%) weak (κ = 0.35) and one (10%) moderate (κ = 0.41) agreement were recorded between observers.

Weak (40%) to moderate (60%) inter-observer agreements were recorded for LCFH between observers. The highest agreement was κ = 0.57, and the lowest was κ = 0.25.

Most of the inter-observer agreements (90%) regarding GDJD and GDAR were below 0.20, indicating poor-to-no agreements. The highest kappa values for GDJD and GDAR were 0.29 and 0.21, respectively. No good or excellent agreement was recorded for any of the qualitative parameters in this study. The results of Cohen’s kappa coefficient for qualitative values are shown in the Appendix A (Appendix A).

## 4. Discussion

This study was designed to assess the inter-observer reliability of the radiographic methods reported for diagnosis of the coxofemoral joint disease in juvenile Rottweilers. The aim of our study was to evaluate the reproducibility of the radiographic examinations by different groups of experienced investigators in a closed cohort of four-month-old Rottweilers. We hypothesized that there would be a good inter-observer agreement for the values measured in this study.

Our findings indicate that the inter-observer agreements among the observers varied between the quantitative and qualitative results. High and low inter-observer agreements were recorded for quantitative and qualitative values, respectively. Therefore, our hypothesis was partially confirmed.

All of the quantitative values (NA, DI, DARS, and CEA) had excellent to good inter-observer agreements in this study (75% excellent and 25% good agreement), whereas the reported results for qualitative values had week to poor agreements. These findings confirm the results of the previously performed studies, which reported low agreement for qualitative values (qualitative parameters of the FCI scoring method) and good inter-observer agreement for quantitative values such as NA and DI [38,39,40,41]. Based on our findings, it can be deduced that quantitative values are more accurate and reliable for diagnosis of the coxofemoral joint diseases. The results of the general linear model confirm these findings, where no significant differences were recorded between observers regarding NA, DI, and CEA (except for one observer) and only a few differences were recorded between observers regarding DARS. Contrary to quantitative values, most of the qualitative measurements had poor agreements. The reason for low agreement of the qualitative values might arise from the nature of these values, which are more relative to the observers than quantitative values; therefore, the distinction of the qualitative parameters might differ between observers, especially in the case of near-normal or mild hip dysplasia [39].

Given that there is no single standard method for the diagnosis of CHD, the accuracy of the investigated values can only be evaluated by intra- and inter-observer agreements [38]. Due to the qualitative nature of most scales for observers and because of the importance of the identification of the anatomic landmarks in measurements, it is important to have a repeatable and reproducible method to assess CHD. The screening protocols with poor intra- and inter-observer agreements represent inaccurate measurements and can cause inefficient screening programs and affect preventive or therapeutic planning; therefore, a combination of different screening protocols such as FCI, OFA, or BVA/KC which are very similar to each other, with other methods such as PennHIP and clinical examinations might increase the accuracy of the results.

An intra- and inter-observer study on the measurement of the femoral and tibial alignments in small-breed dogs showed that, upon increasing the number of the anatomical landmarks or increasing the complexity of the anatomical structures, the intra- and inter-observer agreement decreased [42]. Due to the importance of the early diagnosis of CHD and due to the complexity of the screening protocols, it would be of interest to evaluate the accuracy of the measurements of the hip joint, especially between different observers with high levels of experience.

The experience of the observers can affect the examinations. A study by Verhoeven et al. [39] showed that the inter-observer agreement increases with the experience of the observers. All of the observers in our study were experienced surgeons/radiologists. Observers 1 and 4 were certified CHD scrutineers (observer 1: surgeon; observer 4: radiologist) and observer 2 (orthopedic surgeon) was used to performing hip examinations routinely in the clinic due to the nature of her job. Observer 3 had less experience than the other examiners in this study and, despite the high level of experience of observer 5 in diagnostic imaging, this observer did not perform hip examinations routinely; thus, observers 1, 2, and 4 had more experience in hip examinations than the others.

No significant difference was recorded between surgeons regarding NA, DI, DARS, and CEA. Results of the examinations of the radiologist were similar to those of the surgeons regarding NA, DI, and CEA, and only in one case (CEA) a significant difference (*p* = 0.04) was recorded between one of the surgeons (observer 2) and one of the radiologists (observer 4). The mean difference and standard error between these observers were only 2.9° ± 1°, which might be negligible due to the small amount and low impact on the final results.

The results reported for DARS differed between observers. No significant difference was recorded between more experienced observers (observers 1, 2, and 4), whereas the measurements of observers 3 and 5 differed significantly from others (*p* = 0.00). The mean difference and standard error of these measurements were 7.1° ± 0.9°. This difference might be significant, as the hips with DARS ≤ 7.5° are considered healthy [33]. This result confirms the impact of the experience of the observer in the measurement of complex values. It would be of interest to investigate the agreement between DARS and CEA with the final FCI grade. Due to the complexity of these measurements, and poor results extracted from observers with lower experience, it could be deduced that DARS and CEA may not be appropriate for routine examinations and should be performed by certified CHD scrutineer to increase the accuracy.

The measurement of the qualitative results was similar between all the observers, and poor agreements were recorded. Low intra- and inter-observer reliabilities were reported in previous studies regarding subjective scoring of the hips on ventrodorsal hip-extended radiographs [38,39,40,43]. In an inter-observer study between different observers from different European countries, despite a specified evaluation method (FCI criteria), various results were reported and the agreement was low [39]. These findings emphasize the importance of standardizing the film reading and re-evaluation of the screening protocols.

Despite previous studies that evaluated a combination of different dog breeds, in the current study we investigated inter-observer reliability of the different methods for early diagnosis of the CHD on only one specific dog breed. We presume that the breed difference might be a significant factor in diagnosing CHD, as the results of our previous study [23] on purebred Rottweilers differed from previously reported results in the literature.

Despite the recommended examination age (12 or 18 months) from the FCI, we investigated values in four-month-old dogs, which increased the difficulty of the procedure due to the lack of perfect skeletal maturity. At four months of age, relative musculoskeletal growth is reached; therefore, different studies investigated dogs from this age [44]. It should be noted that because of the relative musculoskeletal maturity, growth, and subsequent increased pressure on the hip joint and therefore increased incidence of the clinical and radiographical signs, the results of the early diagnosis of CHD at older ages are more reliable [23,44]. On the other hand, early detection of the hip laxity in puppies would help veterinarians perform preventive or therapeutic procedures to reduce the severity of the disease in older ages such as juvenile pubic symphysiodesis (JPS), which should be performed on dogs of a young age (3.5–4 months) [45].

Our findings underline the importance of accurate diagnostic methods to achieve accurate results. According to the importance of heredity in controlling CHD, false-negative results would allow the breeding of dysplastic dogs, which are falsely considered as non-dysplastic, and prevent declination of the population of the dysplastic dogs. A retrospective study in Switzerland showed that, over a period of 20 years, the incidence of the mild-to-severe grades of the CHD (C, D, and E) decreased significantly [46]. Similar results were reported in France between 1997 and 2017 for large breeds such as Cane Corso, Gordon Setter, Rottweiler, and White Swiss Shepherd [47]. Therefore, it is recommended to use methods with high inter- and intra-observer agreements, to achieve better results and decrease the incidence of the disease in future.

The image quality of the radiograph is another important topic that could affect the assessments. An inter-observer study showed that improving the technical quality of the radiographs does not improve the inter-observer agreement [40], while positioning errors can affect radiographic examinations [48,49]. Therefore, all of the included radiographs in our study were taken according to the FCI guidelines by experienced technicians and evaluated by experienced radiologists, to confirm the inclusion criteria and to eliminate the risk of positioning errors in this study.

Our study had some limitations. All of the investigations were performed on four-month-old puppies, which made radiographic examinations more difficult due to the immature/newly matured musculoskeletal system of the dogs. The intra-observer agreements were not assessed in this study; these could give us more information about the repeatability of the examinations. In a few cases, some of the qualitative parameters were reported to be indefinable by observer 5 (4.1% of the measurements). These missing data might have a negligible impact on statistics. Investigating a specific breed at a certain age, despite being intentional, makes the comparison between current and previous studies that have used a combination of different breeds at different ages more difficult. Further investigations are needed to evaluate the influence of aging on the inter-observer agreements, as detection of the degenerative changes might be easier in older ages; however, it would be of interest to investigate the final FCI score of the Rottweilers included in this study and assess the predictability of the methods used at four months of age.

## 5. Conclusions

In conclusion, excellent-to-good inter-observer agreements were recorded for quantitative values (NA, DI, DARS, and CEA) in our study, whereas the results of the inter-observer evaluation of the qualitative values (SCAR, LCFH, GDJD, and GDAR) were poor. The results reported by surgeons and radiologists were mostly similar, and only one observer (radiologist) had different results than others. According to the significance of the early diagnosis of CHD, evaluated quantitative values could be considered as reliable values, and a combination of these methods with other methods might increase the accuracy of the examinations and improve the efficiency of the screening methods.

## Figures and Tables

**Figure 1 animals-12-01269-f001:**
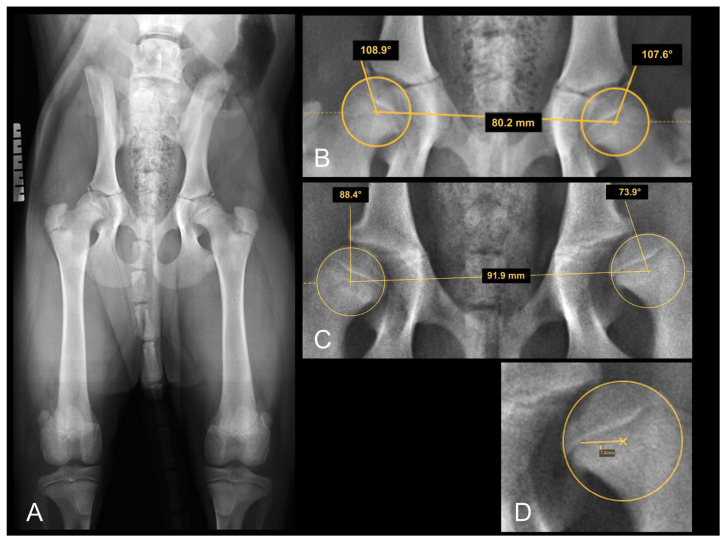
(**A**) Extended FCI conform ventrodorsal radiograph of a four-month-old Rottweiler with what were assumed to be normal hip joints. Note the transitional lumbosacral vertebrae (type III). Right is always to the left of the images. (**B**) Magnification of (**A**), focusing on the hip joints: demonstration of Norberg angle measurement. (**C**) Section of a VD radiograph in another four-month-old Rottweiler puppy with obvious laxity of both hip joints, more severe on the left side. The femoral head center is positioned on, or barely medial to, the dorsal acetabular rim in (**B**) and definitely lateral to the DAR in (**C**) bilaterally. (**D**) Magnification of (**C**), demonstrating the actual distance of the femoral head center from the DAR.

**Figure 2 animals-12-01269-f002:**
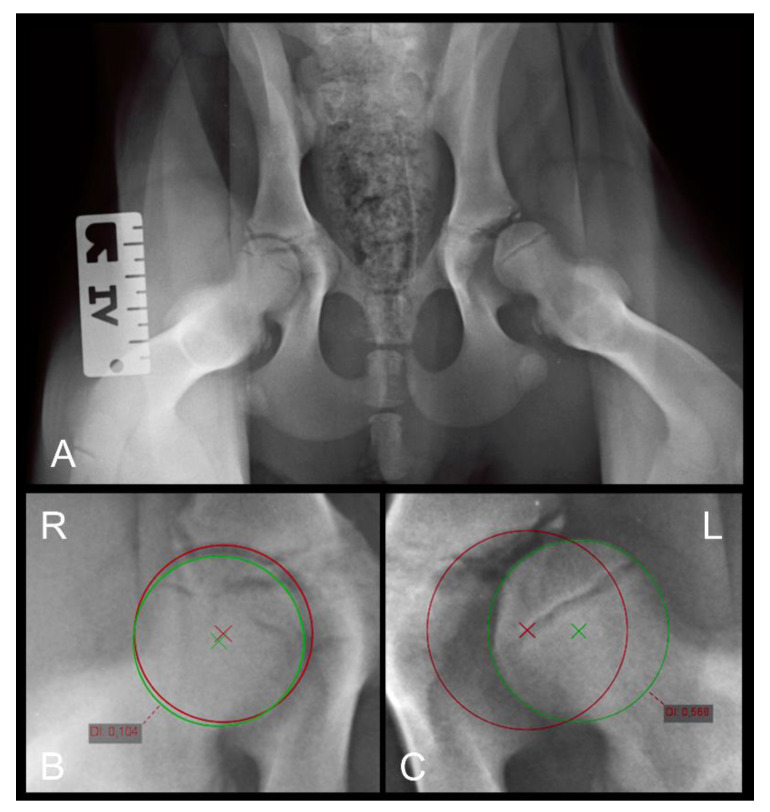
(**A**) Distraction view VD radiograph of a four-month-old Rottweiler (same dog as in Figure 1A,B). (**B**) A magnification of the right and (**C**) a magnification of the left hip joint demonstrating the distraction index measurements. Note the laxity in the left joint which was not obvious in the extended view.

**Figure 3 animals-12-01269-f003:**
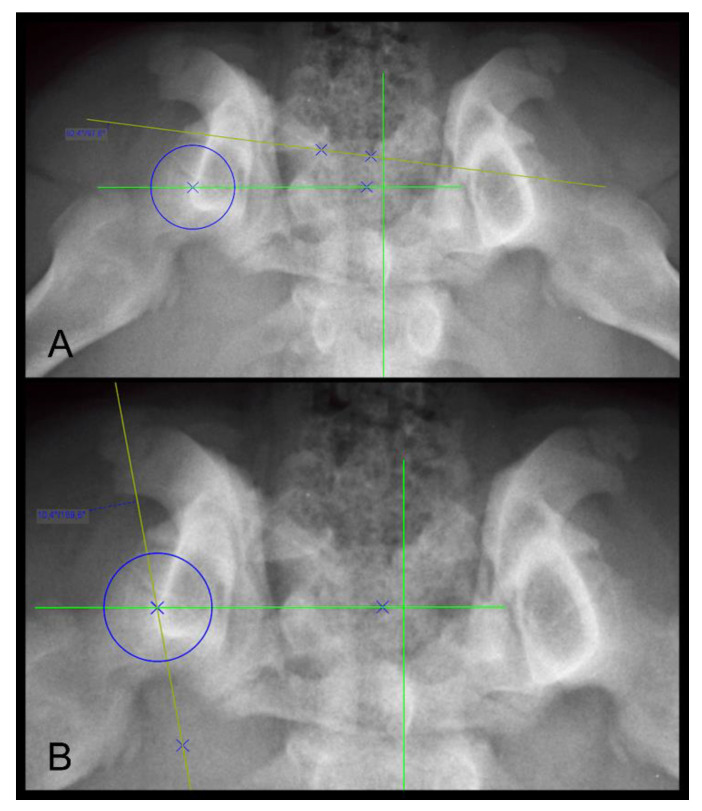
Dorsal acetabular rim view radiographs of the same Rottweiler shown in Figure 1 A,B as well as Figure 2. Measurement of the dorsal acetabular rim slope is demonstrated on (**A**) and of the center edge angle on (**B**). Right is always to the left of the image.

**Table 1 animals-12-01269-t001:** Descriptive statistics of the measured quantitative values in four-month-old Rottweilers.

		NA °	DI ^mm^	DARS °	CEA °
Mean ± SD ^1^	Observer 1	103 ± 7.3	0.3 ± 0.1	7.3 ± 3.5	16.9 ± 4
	Observer 2	100.5 ± 6	0.3 ± 0.1	8.2 ± 5.3	17.8 ± 5.2
	Observer 3	99.4 ± 7.8	0.4 ± 0.2	11.6 ± 4	16 ± 5.6
	Observer 4	102.9 ± 7.2	0.3 ± 0.1	7.7 ± 4.7	14.8 ± 5.7
	Observer 5	103.1 ± 6.5	0.3 ± 0.2	4.4 ± 5.6	16 ± 6
	Overall	101.8 ± 7.1	0.3 ± 0.2	8.1 ± 5	16.2 ± 5.5
Range ^2^		118–72	1–0	33.7–0.1	30.3–1.6

**^1^** Standard Deviation. ^2^ Maximum–minimum.

**Table 2 animals-12-01269-t002:** Percentage (%) of the diagnosed qualitative values by each observer at four months old.

	**SCAR**				
Observer	Regular and Thin Sclerosis		Regular and Thick Sclerosis		Laterally Increased Thickness of Sclerosis
1 (Surgeon) *	100.0		0		0
2 (Surgeon)	76.8		23.2		0
3 (Radiologist)	68.5		16.7		14.8
4 (Radiologist) *	96.4		3.6		0
5 (Radiologist)	70.0		30.0		0
Total	84.1		12.6		3.3
	**LCFH**				
Observer	Medial to DAR		Superimposing DAR	0–2 mm Lateral to DAR	More than 2 mm Lateral to DAR
1 (Surgeon)	53.7		37.0	7.4	1.9
2 (Surgeon)	57.1		19.6	21.4	1.8
3 (Radiologist)	44.6		25.0	23.2	7.1
4 (Radiologist)	41.2		35.3	21.6	2.0
5 (Radiologist)	49.0		29.4	19.6	2.0
Total	49.3		29.1	18.7	3.0
	**GDJD**				
Observer	Congruent Joint Space and No Osteophytes		Incongruent Joint Space and No osteophytes		Incongruent Joint Space with Osteophytes
1 (Surgeon)	88.9		11.1		0
2 (Surgeon)	55.4		44.6		0
3 (Radiologist)	44.6		44.6		10.7
4 (Radiologist)	76.8		21.4		1.8
5 (Radiologist)	75.9		11.1		13.0
Total	68.1		26.8		5.1
	**GDAR**				
Observer	Triangular Shape and No Osteophytes	Mild Rounded DAR and No Osteophytes	Rounded DAR with Mild Osteophytes	Truncated Shape of the DAR with Laxity and Less Roofing	Rounded Shape of the DAR with Severe Osteophytes
1 (Surgeon)	79.6	18.5	0	1.9	0
2 (Surgeon)	12.5	69.6	17.9	0	0
3 (Radiologist)	1.8	41.1	35.7	16.1	5.4
4 (Radiologist)	37.5	41.1	16.1	5.4	0
5 (Radiologist)	28.3	32.1	26.4	7.5	5.7
Total	31.6	40.7	19.3	6.2	2.2

* Certified CHD scrutineer; SCAR, sclerosis of the cranial acetabular rim; LCFH, location of the center of the femoral head; GDJD, grading of the degenerative joint disease; GDAR, grading of the dorsal acetabular rim.

**Table 3 animals-12-01269-t003:** Intra-class correlation coefficient (ICC) for inter-observer variability between the observers.

Parameter	ICC	Significance
NA	0.93	*p* < 0.001
DI	0.95	*p* < 0.001
DARS	0.74	*p* < 0.001
CEA	0.86	*p* < 0.001

## Data Availability

The data presented in this study are available in the Appendix A.

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
