# Peer review of "Inter-Observer Agreement in Radiographic Diagnosis of Coxofemoral Joint Disease in a Closed Cohort of Four-Month-Old Rottweilers"

_animals, 2022, doi:10.3390/ani12101269_

Round 1
Reviewer 1 Report
Dear Authors, I think the paper is well studied, explained and discussed and I think it deserves publication in the present form.
The results are interesting also from a clinical point of view.
The Author have reported results in an easy way, in order to underline the most important informations.
The main question addressed by the research is the inter observer agreement between radiologist and surgeons about quantitative parameters in radiographic diagnosis of coxofemoral Joint Disease. Moreover the paper underline that qualitative parameters are instead no reliable.
I do no consider the topic original but I think it is very relevant to this topic because it underlines the subjectivity of qualitative parameters such as sclerosis of the cranial acetabular rim (SCAR), location of the center of the femoral head (LCFH), grading of the degenerative joint disease (GDJD), and grading of the dorsal acetabular rim (GDAR) that are some of the parameters that clinics often discuss with owners and breeders and that often can create legal issues. It is important probably to discuss with owners and report more about qualitative parameters that it has been demonstrated from this paper have a higher and stronger inter observer agreement.
I think this paper add to the subject area compared with other published material a good population examined, a clear study design and goal, a review of qualitative and quantitative parameters to judge and classify hip dysplasia. It is a review well explained of what and how to judge and discuss preventive hip radiographs, with a simple, clear and very well explained objective of the study.
In some part of the study the result is redundant.
The conclusions are consistent with the evidence and arguments presented and they address the main question posed.
The references are appropriate.
Figures are clear, of good quality and well explained. Tables are easy to understand.
Author Response
Author's Reply to the Review Report (Reviewer 1)
Dear Reviewer,
Thank you very much for your comments. I greatly appreciate the time you spent reading our manuscript.
In response to concerns raised by reviewers during the review process, a minor revision was made. The recommendations made have added clarification and quality to our study and we hope that the revised manuscript is suitable for publication and look forward to hearing from you.
Please see the following letter that addresses the suggestions made. We hope that the revised manuscript is suitable for publication.
Comments:
In some part of the study, the result is redundant.
Response:
Dear reviewer, to make the results more comprehensible, redundant parts (line 249-256 in the old manuscript and line 255 in the revised version) have been deleted, as they were reported in the Quantitative Results (line 257-271) and table 2.
Yours Sincerely,
Corresponding author
Reviewer 2 Report
Thank you to the authors for analyzing the inter-observer agreement in both quantitative and qualitative measurements and characteristics of canine hip dysplasia. The evaluation at a young age is an important question and one I am personally interested in investigating as you astutely state for early therapeutic or preventative interventions. The incorporation of both surgeons and radiologists is a big strength of this paper. Your study and manuscript were well designed and written making it a pleasure to read and review. I have a few points of clarity and suggestions for the manuscript.
Line 64- suggest altering this statement to add in medium to large breed dogs for a screening time of 1 year.
Line 68-69: " dysplastic dogs should be excluded from breeding programs" Although I agree highly dysplastic dogs should not be bred. A quantifying statement of such should be included. Breeding as you know is multifactorial and excluding solely on CHD would decrease the genetic diversity of a breed especially those with historically increased laxity.
Line 118-119: Please clarify what is not related "too closely". This can be misinterpreted as too closely is highly subjective. If possible, provide a generational linkage or a COI statement.
Line 120-121: I suggest adding in "being free of any clinical musculoskeletal diseases related to joints other than hips." Especially since you note a lumbosacral transitional vertebrae in Figure 1.
Line 134 and throughout paper- Specifically in line 134, you mention the Baderstscher method modified by Vezzoni but then subsequently mention the PennHIP measurements method. Please clarify how these measurements were obtained for the distraction view. The PennHIP method calculations are calibrated by using the distraction apparatus used in the PennHIP distraction view. It appears by Figure 2, the distraction apparatus was not used to obtain this view. Therefore, please clarify the reference of measurement for using the PennHIP measurement methodology.
Line 249-250: suggest rewording, the sentence structure is a bit cumbersome and takes away from the important results stated.
Line 266-268: Please clarify how this case was identified, a potential explanation for the significant difference, and if this case was removed from the analysis.
Line 371: Please clarify what it means to be a certified CHD expert?
Line 384-394: Mention of disagreement between observes and experience should be mentioned in the results section of the study. As you state this experience difference may play a role in accuracy of data.
Line 430: Suggest changing sentence to "the image quality of the radiograph" as I would say positional errors would be looped in with the overall quality of the radiograph.
Line 438: Why was intra-observer agreement not studied?
Line 440-441: Please clarify what data was missing or indefinable and add it to the methods sections with a note of impact on statistics if applicable.
Author Response
Author's Reply to the Review Report (Reviewer 2)
Dear Reviewer,
Thank you very much for your comments, critiques, and suggestions for our manuscript. The majority of the recommendations made have added clarification and quality to our study.
We feel that the revised manuscript has addressed the concerns raised during the review process. Please see the following letter that addresses the suggestions made. We hope that the revised manuscript is suitable for publication.
Comments:
- Line 64- suggest altering this statement to add in medium to large breed dogs for a screening time of 1 year.
Response: As suggested, this statement has been revised (line 64-65).
- Line 68-69: " dysplastic dogs should be excluded from breeding programs" Although I agree highly dysplastic dogs should not be bred. A quantifying statement of such should be included. Breeding as you know is multifactorial and excluding solely on CHD would decrease the genetic diversity of a breed especially those with historically increased laxity.
Response: As suggested, this part has been revised (line 70-72).
- Line 118-119: Please clarify what is not related "too closely". This can be misinterpreted as too closely is highly subjective. If possible, provide a generational linkage or a COI statement.
Response: Dear reviewer, following the evaluation of the pedigree data of the dogs, it has been proven that none of the dogs had common parents. To avoid misinterpretations this sentence has been revised (line 121-122).
- Line 120-121: I suggest adding in "being free of any clinicalmusculoskeletal diseases related to joints other than hips." Especially since you note a lumbosacral transitional vertebrae in Figure 1.
Response: As suggested, this part has been revised and the word "clinical" has been added (line 123-124).
- Line 134 and throughout paper- Specifically in line 134, you mention the Baderstscher method modified by Vezzoni but then subsequently mention the PennHIP measurements method. Please clarify how these measurements were obtained for the distraction view. The PennHIP method calculations are calibrated by using the distraction apparatus used in the PennHIP distraction view. It appears by Figure 2, the distraction apparatus was not used to obtain this view. Therefore, please clarify the reference of measurement for using the PennHIP measurement methodology.
Response: Dear reviewer, the radiographic positioning of the dogs in distraction view was done according to the Badertscher method modified by Vezzoni (with distraction apparatus in all dogs), and the calculation of the distraction index was done according to the PennHIP measurement method (DI= distance between the centers of the femoral head and acetabulum/ the radius of the femoral head during distraction). As a clarification, a statement has been added to the DI part (line 193-195).
- Line 249-250: suggest rewording, the sentence structure is a bit cumbersome and takes away from the important results stated.
Response: Dear reviewer, to make the results more comprehensible, redundant parts (line 249-256) have been deleted, as they were repeated in the quantitative results and table 2.
- Line 266-268: Please clarify how this case was identified, a potential explanation for the significant difference, and if this case was removed from the analysis.
Response: Dear Reviewer, as mentioned in the material and methods (line 232-236), the differences between observers were analyzed using a general linear model (GLM) and multiple comparisons were performed using Bonferroni's alpha correction procedure. The results with a p-value < 0.05 were considered significant. You can find the comparisons in the supplementary material. In this case (CEA measurements between observers) only a significant difference has been recorded between observers 2 and 4 (p-value = 0.04) with 2.9° ± 1° standard error. This result was not removed from the analysis, as we obtained these results after statistical analysis. Furthermore, these comparisons have been done between all CEA measurements of observer 2 and all CEA measurements of observer 4 and not only between two single measurements, thus it is not possible to exclude them from our analysis.
- Line 371: Please clarify what it means to be a certified CHD expert?
Response: we have meant certified scrutineers for canine hip dysplasia here. The following parts have been revised to clarify the sentences and "certified CHD experts" has been changed to "certified CHD scrutineers" (line 147, 300, 368, 390-391).
- Line 384-394: Mention of disagreement between observes and experience should be mentioned in the results section of the study. As you state this experience difference may play a role in accuracy of data.
Response: As recommended, the disagreement between observes and experience have been mentioned in the results (line 261-263), and the interpretation of this disagreement has been left in the discussion (line 383-385).
- Line 430: Suggest changing sentence to "the image quality of the radiograph" as I would say positional errors would be looped in with the overall quality of the radiograph.
Response: As suggested, this sentence has been revised (line 428), furthermore, the word "technical" has been added to line 429 to clarify the sentence.
- Line 438: Why was intra-observer agreement not studied?
Response: Due to the high number of the study cases, and the high workload of the observers. The intra-observer study (second round of the measurements) did not perform.
- Line 440-441: Please clarify what data was missing or indefinable and add it to the methods sections with a note of impact on statistics if applicable.
Response: As mentioned in the text, a few numbers of the qualitative values were reported to be indefinable by observer 5. As suggested, a revision has been done (line 439-441), furthermore a statement has been added to the materials and methods (line 249-250).
At the end, I greatly appreciate the time you spent reading our manuscript and look forward to hearing from you.
Yours Sincerely,
corresponding author
Reviewer 3 Report
The inclusion criteria were being free of any musculoskeletal diseases related to the joints other than hips. How many dogs in the study had clinical symptoms of hip dysplasia???? Why they do not report in the article the findings of the clinical examination??
Author Response
Author's Reply to the Review Report (Reviewer 3)
Dear Reviewer,
Thank you for your comments. I greatly appreciate the time you spent reading our manuscript.
In response to concerns raised by reviewers during the review process, a minor revision was made. The recommendations made have added clarification and quality to our study. Please see the following letter that addresses the suggestions made. We hope that the revised manuscript is suitable for publication and look forward to hearing from you.
Comments for Authors:
The inclusion criteria were being free of any musculoskeletal diseases related to the joints other than hips. How many dogs in the study had clinical symptoms of hip dysplasia???? Why they do not report in the article the findings of the clinical examination??
Response:
Dear Reviewer, the inclusion criteria in our study were being free of any musculoskeletal diseases related to the joints other than hips but all of the included dogs in our study were clinically sound dogs without any clinical musculoskeletal diseases. Thus, no dog had a symptom of hip dysplasia. To make it more clear the sentence "related to the joints other than hips" has been deleted. Please see line (123-124).
Yours Sincerely,
Corresponding author